Efficacy and safety of intracavitary electrocardiography-guided peripherally inserted central catheters in pediatric patients: a systematic review and meta-analysis

Zhang Li zhangli202406@163.com
Wang Min
Zhao Mingjia
Pu Siyi
Zhao Jiao
Zhu Ge
Zhang Qin
Li Dan
Nanchong Central Hospital , Nanchong , China
Menini Stefano
Electronic publication date: 2024 Oct 8
Publication date: 2024
Volume: 12
Electronic Location ID: e18274
Received 2024 Jun 17; Accepted 2024 Sep 18
Copyright: © 2024 Zhang et al.
Copyright year: 2024
Copyright holder: Zhang et al.
License: This is an open access article distributed under the terms of the Creative Commons Attribution License, which permits unrestricted use, distribution, reproduction and adaptation in any medium and for any purpose provided that it is properly attributed. For attribution, the original author(s), title, publication source (PeerJ) and either DOI or URL of the article must be cited.
License URL: https://creativecommons.org/licenses/by/4.0/

Keywords: Intracavitary electrocardiography, Peripherally inserted central catheter, Children, Treatment outcome, Neonate

Funding: The authors received no funding for this work.

==============================
Objective

To assess the efficacy and safety of intracavitary electrocardiography (IC-ECG)-guided peripherally inserted central catheter (PICC) placements in pediatric patients, emphasizing improvements over traditional placement methods.

Methods

A literature search was conducted in April 2024 across PubMed, Cochrane Library, and EMBASE. Studies focusing on pediatric patients and reporting the efficacy and safety of IC-ECG-guided PICC placement via the upper extremity were included. This study was registered with the PROSPERO database (CRD42024549037) in accordance with PRISMA guidelines.

Results

Eleven studies were included, comprising five randomized controlled trials (RCTs) and six quasi-experimental studies. The pooled analysis showed that IC-ECG had an applicability and feasibility of 97% and 98%, respectively. The first puncture success rate was 91%, and the overall success rate was 98%. Sensitivity and specificity were 97% and 80%, respectively. IC-ECG significantly reduced overall complications compared to traditional methods (RR: 0.31, 95% CI [0.20–0.46], p < 0.00001), particularly in phlebitis (RR: 0.25, 95% CI [0.11–0.57], p = 0.001) and arrhythmias (RR: 0.09, 95% CI [0.01–0.70], p = 0.021). Similar results were observed in neonates. Only one case of catheter-related bloodstream infection (CRBSI) was reported, and no arrhythmia events were noted.

Conclusion

IC-ECG-guided PICC placement is a highly effective and safe method for pediatric patients, including neonates, offering significant advantages over traditional techniques. Further high-quality studies are needed to standardize procedural techniques and explore cost-effectiveness.

Introduction

Peripherally inserted central catheters (PICCs) are widely used in neonatal intensive care units (NICUs) due to their ability to provide long-term venous access for critically ill newborns. This includes low birth weight and extremely low birth weight infants, children with malignant tumors, and other critically ill patients (Johansson et al., 2013; Johnson et al., 2016; Li et al., 2019; Ozkiraz et al., 2013). These catheters are crucial for the administration of high-viscosity fluids or irritant medications, such as those used in chemotherapy, nutrition, and other intensive treatments that peripheral veins cannot handle (Janes et al., 2000; Mielke, Wittig & Teichgraber, 2020). Proper placement of the PICC tip is essential for minimizing complications and ensuring effective treatment. The optimal position for the PICC tip is at the cavoatrial junction (CAJ), where the superior vena cava (SVC) meets the right atrium (RA) (Gorski et al., 2021; Westergaard, Classen & Walther-Larsen, 2013). Incorrect placement can lead to several complications. If the tip is too high, it increases the risk of venous thrombosis and catheter occlusion. If it is too low, there is a risk of cardiac tamponade, arrhythmias, and perforation of the heart or great vessels (Ares & Hunter, 2017; Jain, Deshpande & Shah, 2013). These risks are particularly significant in neonates and young children due to their smaller anatomical structures and the ongoing development of their organ systems, which differ considerably from adults (Inagawa et al., 2007). For example, the anatomical landmarks used to guide PICC placement in adults are not always reliable in neonates, making the process more challenging and the need for accurate, real-time verification methods more critical. Traditional methods, such as post-procedural chest radiography, are less effective due to these anatomical differences and pose additional risks from radiation exposure (Avni, Greenspan & Goldberger, 2011; Xiao et al., 2020; Zhou et al., 2017b).

Intracavitary electrocardiography (IC-ECG) is a technique initially reported in 1949 for guiding the placement of PICCs by using the catheter as an intracavitary electrode to monitor P-wave changes on an electrocardiogram. When the catheter tip reaches the CAJ, the P-wave amplitude peaks, indicating optimal positioning (Hellerstein, Pritchard & Lewis, 1949; Jeon et al., 2006). In adult patients, IC-ECG has proven to be a highly accurate, cost-effective method, reducing the need for post-procedural chest radiographs and minimizing radiation exposure. This technique effectively decreases the risk of complications associated with improper catheter tip placement, such as venous thrombosis, cardiac tamponade, and arrhythmias (Liu et al., 2019; Yuan et al., 2017). Given its success in adults, IC-ECG has been adapted for use in pediatric patients, including neonates (Pittiruti et al., 2024; Rossetti et al., 2015; Zhou et al., 2017b). Studies have demonstrated that IC-ECG-guided PICC placement in children is highly effective, significantly improving the success rate of placements, reducing procedural times, and lowering the incidence of complications (Capasso et al., 2018; Xiao et al., 2020; Zhu et al., 2021). There is currently a lack of systematic evidence regarding the use of IC-ECG-guided PICC placement in children. Therefore, this study aims to systematically evaluate the efficacy and safety of IC-ECG-guided PICC placement in pediatric patients. Additionally, we conduct a meta-analysis of randomized controlled trials (RCTs) comparing IC-ECG guidance with traditional placement methods.

Materials and Methods

This systematic review and meta-analysis strictly adhere to the Preferred Reporting Items for Systematic Reviews and Meta-Analyses (PRISMA) guidelines and have been registered with the PROSPERO database (University of York, UK) under the identifier CRD42024549037.

Literature search

A comprehensive literature search was conducted in April 2024 across databases including PubMed, Cochrane Library, and EMBASE. Search terms used were ‘electrocardiography,’ ‘peripherally inserted central catheter,’ and ‘children.’ Detailed search strategies are provided in Table S1. The search was confined to English-language studies published from the year 2000 onwards. Additionally, relevant references from identified articles were manually searched through PubMed to ensure a thorough collection of pertinent studies.

The literature search was independently conducted by M.W. and MJ.Z. In cases of disagreement, discussions were facilitated and conclusively arbitrated by the senior author, L.Z, who served as the final referee for resolving any inconsistencies.

Inclusion and exclusion criteria

Studies were selected based on the following criteria: inclusion criteria were studies focusing on pediatric patients (neonates, infants, and children), reporting the efficacy and safety of IC-ECG-guided PICC placement via the upper extremity, and including RCTs, non-randomized controlled trials (non-RCTs), and observational studies (including cohort and case-control studies). Exclusion criteria were studies not centered on IC-ECG guidance for PICC placement, studies without relevant data on the efficacy and safety of the intervention, studies not involving pediatric patients, publications categorized as letters, case reports, commentaries, conference abstracts, and studies with incomplete data or irrelevant outcomes.

Quality assessment

The quality of included RCTs was assessed using the Jadad scale, with scores between four and seven indicating high quality (Jadad et al., 1996). Bias risk in RCTs was evaluated using the Cochrane Risk of Bias tool. Quasi-experimental studies were assessed using the Joanna Briggs Institute (JBI) critical appraisal tools (Barker et al., 2024). Each study underwent independent quality and bias risk assessment to ensure the robustness of the meta-analysis.

Data extraction

Data extraction involved a comprehensive review of key elements from each included study. Extracted data included authors, publication year, study design, participants, sample size, and specific preoperative data such as gestational age (weeks), age (months), male-to-female ratio, and age at first catheterization (days). Outcome metrics analyzed were applicability (proportion of children with identifiable and changing P wave amplitude during PICC advancement), feasibility (percentage of cases with clear identification of the peak P wave corresponding to the passage between the SVC and RA), accuracy (ratio consistent with X-ray), success rate of first puncture (defined as the successful placement of the PICC tip in the optimal position at the CAJ on the first attempt), overall success rate, sensitivity, specificity, overall complications, and specific adverse events such as catheter site inflammation, phlebitis, catheter-related bloodstream infection (CRBSI), thrombosis, catheter malposition, and arrhythmia.

Statistical analysis

Statistical analyses were conducted using Stata software, Version 17. Dichotomous outcomes were evaluated through relative risks (RRs), reported with 95% confidence intervals (CIs). Heterogeneity among studies was assessed using chi-squared and I-squared (I²) tests. In instances where significant heterogeneity was identified (I² > 50%), random-effects models were applied, and sensitivity analyses were performed to examine the robustness of the findings and identify potential sources of heterogeneity. Fixed-effects models were used for cases of low heterogeneity. A subgroup analysis was conducted specifically on neonatal outcomes to evaluate the efficacy and safety of IC-ECG in this population. Statistical significance was defined as a p-value less than 0.05.

Results

Following a rigorous screening process that involved removing duplicate entries, eliminating irrelevant records, and excluding studies with substandard methodologies or inadequate data, a total of 11 (Capasso et al., 2018; D’Andrea et al., 2023; Gao et al., 2024; Ling et al., 2019; Raffaele et al., 2021; Tang et al., 2021; Xiao et al., 2020; Yang et al., 2019; Zhang et al., 2022; Zhou et al., 2017a; Zhu et al., 2021) studies were selected for our systematic review, including five RCTs (Gao et al., 2024; Ling et al., 2019; Tang et al., 2021; Xiao et al., 2020; Zhu et al., 2021) and six quasi-experimental studies (Capasso et al., 2018; D’Andrea et al., 2023; Raffaele et al., 2021; Yang et al., 2019; Zhang et al., 2022; Zhou et al., 2017a). A meta-analysis was conducted on the five RCTs. This thorough filtering process is illustrated in Fig. 1. Table 1 provides a systematic compilation of the baseline characteristics and quality evaluations of these studies. Risk of bias was assessed using the Cochrane Risk of Bias tool, revealing that among the five RCTs, one was at low risk and the remaining four were at moderate risk, as shown in Fig. 2. The risk assessment of the six quasi-experimental studies using the JBI critical appraisal tools indicated that all could be included in the analysis, with detailed results in Table S2.

Figure 1 PRISMA flow diagram.

Table 1 Characteristics and quality assessment of included studies.

Author, year	Study design	Participants	No. of patients	Gestational age or age	Male: Female	Age for first catheterism (days)	Quality scored	
Capasso et al. (2018)	Q	Neonates	39	29.41 ± 3.04a
(weeks)	30:9	12.3 ± 3.58	NA	
D’Andrea et al. (2023)	Q	Neonates	105	32 (26–37)b
(weeks)	NA	16 (7–67)b	NA	
Gao et al. (2024)	RCT	Infants	90/90	43.24 ± 0.47/41.36 ± 0.38
(mons)	51:39/53:37	NA	4	
Ling et al. (2019)	RCT	Neonates	80/80	37.1 ± 1.4/36.8 ± 1.3
(weeks)	46:34/48:32	3.4 ± 0.5/3.2 ± 0.3	5	
Raffaele et al. (2021)	Q	Children	31	NA	NA	NA	NA	
Tang et al. (2021)	RCT	Neonates	105/105	36.9 ± 1.5/16.4 ± 1.6
(weeks)	54:51/58:47	3 ± 0.8/2.9 ± 0.8	5	
Xiao et al. (2020)	RCT	Neonates	78/83	32.17 ± 2.63/32.36 ± 2.78
(weeks)	42:36/43:40	15.21 ± 7.52/13.19 ± 8.8	5	
Yang et al. (2019)	Q	Neonates	173	32.55 ± 2.3
(weeks)	104:69	3.01 ± 4.01	NA	
Zhang et al. (2022)	Q	Children	62	19.1 ± 8.82
(mons)	32:30	NA	NA	
Zhou et al. (2017a)	Q	Infants	49	35 ± 4/36 ± 3
(weeks)	32:17	17 ± 16/13 ± 12	NA	
Zhu et al. (2021)	RCT	Infants	53/53	4 (2–7)/5(3–8.5)c
(days)	29:24/29:24	NA	4	
Notes:

Q, quasi-experimental studies; RCT, randomized controlled trials; NA, not available or not applicable.

a Mean ± Standard Deviation, SD.

b Median (Interquartile range, IQR).

c Median (range).

d Using Jadad scoring rules.

Figure 2 The risk of bias for the included studies according to the Cochrane Risk of Bias tool (Gao et al., 2024; Ling et al., 2019; Tang et al., 2021; Xiao et al., 2020; Zhu et al., 2021).

Our pooled analysis revealed that the applicability of IC-ECG-guided PICC placement in pediatric patients was 97% (91% to 100%), and the feasibility was 98% (95% to 100%). The accuracy of puncture results was 97% (92% to 100%). For pediatric patients, the first puncture success rate was 91% (69% to 100%) and the overall success rate was 98% (93% to 100%). Additionally, the sensitivity and specificity of IC-ECG-guided PICC placement were 97% (94% to 100%) and 80% (43% to 100%), respectively. However, in neonates, the first puncture success rate was slightly lower at 87% (61% to 100%). The sensitivity was 98% (94% to 100%), but the specificity was lower at 63% (39% to 85%).

Regarding complications for pediatric patients, the overall complication rate for IC-ECG-guided PICC placement was 4% (1% to 8%). Specifically, the rates of catheter site inflammation, phlebitis, thrombosis, and catheter malposition were 2% (0 to 7%), 2% (1% to 4%), 1% (0 to 5%), and 4% (1% to 9%), respectively. Detailed results are provided in Table S3. Furthermore, only one case of CRBSI was reported, and no arrhythmia events were noted in all included studies.

Meta-analysis results

Five RCTs compared IC-ECG-guided PICC placement with traditional anatomical landmark-guided PICC placement. The meta-analysis showed that the success rate of the first puncture was significantly higher in the IC-ECG group compared to the traditional group (RR: 1.38, 95% CI [1.06–1.81], p = 0.017, Fig. S1), albeit with high heterogeneity. Sensitivity analysis identified Gao et al.’s (2024) study as the source of heterogeneity (Fig. S2). After excluding this study, the re-analysis confirmed that the success rate of the first puncture remained significantly higher in the IC-ECG group (RR: 1.20, 95% CI [1.10–1.30], p < 0.0001, Fig. 3). Regarding overall complications, the IC-ECG group had a significantly lower incidence compared to the traditional group (RR: 0.31, 95% CI [0.20–0.46], p < 0.00001, Fig. 3). Specific adverse events showed statistically significant differences in the rates of phlebitis (RR: 0.25, 95% CI [0.11–0.57], p = 0.001, Fig. 3) and arrhythmia (RR: 0.09, 95% CI [0.01–0.70], p = 0.021, Fig. 3), whereas there were no significant differences in catheter site inflammation (RR: 0.33, 95% CI [0.07–1.63], p = 0.175, Fig. 4), CRBSI (RR: 0.34, 95% CI [0.07–1.69], p = 0.188, Fig. 4), thrombosis (RR: 0.47, 95% CI [0.21–1.06], p = 0.070, Fig. 4), and catheter malposition (RR: 0.47, 95% CI [0.20–1.13], p = 0.092, Fig. 4).

Figure 3 Forest plot and meta-analysis of the success rate of the first puncture, overall complications phlebitis and arrhythmia (Gao et al., 2024; Ling et al., 2019; Tang et al., 2021; Xiao et al., 2020).

Figure 4 Forest plot and meta-analysis of catheter site inflammation, CRBSI, thrombosis and catheter malposition (Gao et al., 2024; Ling et al., 2019; Tang et al., 2021; Xiao et al., 2020).

Subgroup analysis results of neonates

A subgroup analysis was conducted on neonates based on three included RCTs. The results showed no statistically significant differences between the IC-ECG group and the traditional group in terms of CRBSI (RR: 0.33, 95% CI [0.03–3.17], p = 0.339), catheter site inflammation (RR: 0.33, 95% CI [0.07–1.63], p = 0.175), and thrombosis (RR: 0.25, 95% CI [0.05–1.15], p = 0.076). However, the IC-ECG group demonstrated superior outcomes compared to the traditional group with respect to overall complications (RR: 0.17, 95% CI [0.08–0.36], p < 0.0001), phlebitis (RR: 0.19, 95% CI [0.06–0.63], p = 0.007), and arrhythmias (RR: 0.09, 95% CI [0.01–0.70], p = 0.021). All results are presented in Fig. S3.

Discussion

In the past few decades, postoperative chest X-ray and intraoperative fluoroscopy have been considered the gold standards for evaluating the tip position of central venous access devices, including PICCs. However, these methods have significant drawbacks, such as exposure to ionizing radiation, high costs, and the need for specialized environments like interventional radiology or operating rooms (Keller et al., 2019; Pittiruti et al., 2008). Postoperative chest X-ray clarity can be affected by factors such as central line positioning and projection angle, leading to delays in the use of the catheter and the need for multiple adjustments. This increases the risk of complications, radiation exposure, and infection (Furlong-Dillard, Aljabari & Hirshberg, 2020; Gao et al., 2024). Particularly for children, anatomical differences compared to adults make surface estimations of insertion length unreliable (Ling et al., 2019). Moreover, young children are less cooperative and more prone to crying, leading to higher chest pressure and an increased likelihood of catheter malposition (Zhang et al., 2022). These children cannot independently cooperate with radiologists for chest X-rays. Unlike ultrasound, which requires specialized training, IC-ECG is particularly advantageous due to its ease of learning, operator independence, and high accuracy (D’Andrea et al., 2023). Despite the broad recognition of IC-ECG’s effectiveness and safety in adults, its use in pediatric patients, especially neonates, remains controversial. Therefore, this study aims to systematically evaluate the efficacy and safety of IC-ECG-guided PICC placement in pediatric patients. Given that P waves and QRS waves differ in the lower limbs compared to the upper limbs, and Zhou et al. (2017b) found significant differences in accuracy between upper and lower limb insertions (Ling et al., 2019), we included only studies reporting IC-ECG-guided PICC insertions via the upper limbs.

The single-arm meta-analysis revealed that the applicability and feasibility of IC-ECG were as high as 97% and 98%, respectively. D’Andrea et al. (2023) noted that applicability and feasibility are influenced by the catheter’s bore size and the presence of abnormal rhythms. Capasso et al. (2018) reported that the use of small-caliber epicutaneous cava catheters (ECCs) in neonates, when filled with saline and used with intracavitary electrodes, often impedes the detection of P waves, supporting this viewpoint. Although they suggested that filling ECCs with a hypertonic solution (4%) instead of saline (0.9%) could facilitate P wave observation, most current IC-ECG studies have excluded these catheters. Additionally, Nowlen et al. (2002) reported that a 10% sodium chloride injection is conductive. When this solution is directly injected into the right atrium via the PICC, it can detect changes in the atrial ECG P wave, determining the catheter tip position (McCay, Elliott & Walden, 2014). However, due to the immature renal function of neonates and their limited ability to excrete sodium, a 10% sodium chloride injection is unsuitable for neonates (Yang et al., 2019). Instead, Yang et al. (2019) used 0.9% sodium chloride to assist in positioning neonatal PICC catheters. Among 173 patients, 157 showed characteristic changes in the P wave, indicating no clear consensus on the optimal concentration of the filling fluid for neonatal catheters. Further studies are needed to explore the best concentration gradient. Additionally, the P wave is strongly influenced by patient posture (upright vs. supine), breathing, arm movement, and other factors (Andrew et al., 1994). According to the 2016 American practice standards for infusion therapy, changes in breathing, arm movement, and body position can cause the catheter tip to move up to 2 cm (Zhang et al., 2022). Baseline diseases can also affect the P wave, with conditions like intrauterine distress and apnea potentially causing abnormalities or disappearance of the P wave (Zhu et al., 2021). Finally, the induction of a qualified P wave may be affected by technical factors, including electrode position, voltage, and monitoring system choice, as well as interference from other electronic medical devices (Pittiruti, La Greca & Scoppettuolo, 2011; Rossetti et al., 2015).

The single-arm meta-analysis also found that IC-ECG-guided PICC placement had a sensitivity and specificity of 97% and 80%, respectively. This high accuracy indicates that IC-ECG is highly effective in reducing the need for repositioning the catheter. Capasso et al. (2018) reported only one false-positive case, where the catheter looped in the right atrium, which can be avoided by ensuring that the insertion distance aligns closely with external measurements based on anthropometric data. False negatives were more common, potentially due to mechanical ventilation, electrical interference, or the need to halt the procedure for clinical complications (Capasso et al., 2018; Yang et al., 2019). The difficulty in maintaining a clear ECG signal due to thin neonatal blood vessels and improper catheter movement can affect the stability and clarity of ECG recordings (Ling et al., 2019).

On the other hand, our meta-analysis revealed that the first-attempt success rate for PICC insertions using IC-ECG was higher than that of the traditional method, although with notable heterogeneity. Sensitivity analysis identified the primary source of this heterogeneity as stemming from the study by Gao et al. (2024) which involved older pediatric patients undergoing chemotherapy for malignant tumors. This demographic discrepancy likely contributed significantly to the observed variability. We hypothesize that the improved first-attempt success rate could be indirectly linked to more precise real-time tip positioning, which allows operators to make immediate adjustments, thereby enhancing operator confidence and precision during insertion. Additionally, a multivariable analysis by Zhou et al. (2017b) suggested a correlation between neonatal weight and procedural accuracy, supporting our hypothesis. Furthermore, our aggregated results demonstrated a total complication rate of 4% for the IC-ECG group, markedly lower than that of traditional PICC methods, particularly regarding phlebitis and arrhythmias. Similar results were observed in the neonatal subgroup. This is primarily because IC-ECG facilitates the early and timely detection of potentially misplaced catheters, allowing for immediate corrections. Consequently, this reduces the incidence of catheter malposition, diminishes the need for repeated catheter repositioning or replacements, and minimizes the risk of chemical and mechanical irritation to the fragile vasculature of neonates, thereby reducing infection risks. Moreover, the decrease in malposition rates implies a reduced likelihood of the PICC tip residing in areas of slow blood flow, which lowers the risks of thrombosis and mechanical phlebitis (Ling et al., 2019). Moreover, studies have shown that catheter diameter influences complications. Interestingly, Yu et al. (2023) identified smaller catheter lumens as a major risk factor for PICC tip malposition and occlusion, while Bahl et al. (2023) found that smaller catheters reduce the risk of deep vein thrombosis and are more cost-effective (Yu et al., 2023). This suggests that a balance must be struck when selecting catheter size. Unfortunately, all the RCTs included in our analysis used 1.9 Fr catheters, highlighting the need for further research to explore this issue. In addition, the literature reviewed reported only one case of CRBSI in the IC-ECG group, significantly lower than traditional methods. This finding is notably different from previous studies where CRBSI was the most common PICC-related complication in neonates (Hsu et al., 2010). One possible reason for this discrepancy is that femoral and other lower limb veins, closer to the perineum, have a higher risk of contamination and infection, whereas all the PICC insertions included in our review were performed in upper limbs (Ling et al., 2019). Furthermore, advances in medical practice and improvements in the management of PICC insertions have significantly shortened the indwelling time of catheters compared to earlier practices (Milstone et al., 2013). There is substantial evidence that within 24 h after catheter insertion, a biofilm forms inside the catheter lumen, while a fibrin sheath develops around the outside of the catheter due to its presence as a foreign body. The simultaneous presence of both the biofilm and the fibrin sheath increases the risk of bacterial colonization and infection over time if the catheter remains in place (Tang et al., 2021). Additionally, no cases of arrhythmia were reported in the IC-ECG group, possibly because the operators could continuously monitor the position of the catheter tip as it entered the right atrium, thus avoiding areas prone to triggering arrhythmias, such as the tricuspid valve plane (Rossetti et al., 2015).

Moreover, two studies highlighted that the primary benefit of IC-ECG is its ability to reduce the frequency of adjustments and repeated X-ray positioning, thereby significantly lowering both the duration and cost of procedures compared to traditional methods (Gao et al., 2024; Zhu et al., 2021). However, due to the lack of sufficient data, these outcomes were not subjected to meta-analysis, underscoring the need for further high-quality research in this area. Despite this, our systematic review has demonstrated the accuracy and safety of IC-ECG-guided PICC placement in pediatric settings. It is important to note, however, that IC-ECG has its contraindications, such as in children with myocardial pathology or atrial fibrillation, where it becomes inapplicable. In such cases, Transthoracic echocardiography may be considered as an alternative (Raffaele et al., 2021; Salmela & Aromaa, 1993).

Our systematic review and meta-analysis have several limitations that should be considered. Firstly, although we only included studies with PICC insertions through the upper limbs, which are more reliable for IC-ECG-guided placements, the variability in study designs, patient populations, and procedural techniques across the included studies introduces heterogeneity that may affect the generalizability of our findings. Secondly, the studies included in our analysis were of varying quality, with some studies having a higher risk of bias. While we performed rigorous quality assessments and sensitivity analyses to address this, the potential for residual confounding cannot be entirely eliminated. Thirdly, the number of RCTs included in our meta-analysis was relatively small. Although RCTs provide high-quality evidence, the limited number available may affect the robustness of our conclusions. Additionally, the inclusion of observational studies, while providing a broader evidence base, inherently carries a higher risk of bias compared to RCTs.

Conclusions

IC-ECG-guided PICC placement is an effective and safe technique for pediatric patients, including neonates, significantly improving first-attempt success rates and reducing complications compared to traditional methods. Our meta-analysis demonstrates that IC-ECG has high applicability and feasibility, with a sensitivity and specificity of 97% and 80%, respectively. It effectively reduces the need for repeated adjustments and minimizes the risk of complications such as phlebitis and arrhythmias. Despite these promising results, further high-quality studies are needed to address the observed heterogeneity and validate these findings across diverse clinical settings.

Supplemental Information

Supplemental Information 1 Forest plot and meta-analysis for the success rate of the first puncture with high heterogeneity.

Supplemental Information 2 Sensitivity analysis for first puncture success rate.

Supplemental Information 3 Detailed search strategy.

Supplemental Information 4 Forest plot and meta-analysis of CRBSI, overall complications, catheter site inflammation, phlebitis, arrhythmia, and thrombosis.

Supplemental Information 5 Detailed risk sssessment results for quasi-experimental studies using the Joanna Briggs Institute (JBI) critical appraisal tools.

Supplemental Information 6 Detailed statistical results.

Supplemental Information 7 PRISMA Checklist.

Supplemental Information 8 Rationale and Contribution to Knowledge.

Additional Information and Declarations

Competing Interests

Author Contributions

Data Availability

The authors declare that they have no competing interests.

Li Zhang conceived and designed the experiments, performed the experiments, authored or reviewed drafts of the article, and approved the final draft.

Min Wang conceived and designed the experiments, performed the experiments, authored or reviewed drafts of the article, and approved the final draft.

Mingjia Zhao conceived and designed the experiments, performed the experiments, analyzed the data, authored or reviewed drafts of the article, and approved the final draft.

Siyi Pu performed the experiments, analyzed the data, prepared figures and/or tables, and approved the final draft.

Jiao Zhao performed the experiments, analyzed the data, prepared figures and/or tables, and approved the final draft.

Ge Zhu analyzed the data, prepared figures and/or tables, and approved the final draft.

Qin Zhang analyzed the data, prepared figures and/or tables, and approved the final draft.

Dan Li analyzed the data, prepared figures and/or tables, and approved the final draft.

The following information was supplied regarding data availability:

This is a systematic review/meta-analysis.

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
