# Peer review of "Efficacy and safety of intracavitary electrocardiography-guided peripherally inserted central catheters in pediatric patients: a systematic review and meta-analysis"

_PeerJ, doi:10.7717/peerj.18274_

## Round 0.1 · original submission · Major Revisions

Dear Dr. Zhang,

Your manuscript entitled “Efficacy and Safety of Intracavitary Electrocardiography-Guided Peripherally Inserted Central Catheters in Pediatric Patients: A Systematic Review and Meta-Analysis" which you submitted to PeerJ, has been reviewed by the editor and 2 external reviewers.

The reviewers have raised significant concerns that must be addressed before the manuscript can be considered further. However, since the reviewers see merit in your work, I am open to reconsidering the manuscript if you choose to undertake the suggested revisions and resubmit.

If you decide to resubmit the revised version, please summarize all the improvements made in the new version and give answers to all critical points raised in the reviewers’ report in an accompanying letter. Copy and paste each and every reviewer's comment above your response.

Please note that resubmitting your manuscript does not guarantee eventual acceptance. The revised manuscript will undergo a second round of review by the same reviewers. I must emphasize that the acceptability of the revision will depend upon the resolution of the points raised by the reviewers.

Sincerely yours,
Stefano Menini

·

Basic reporting

Zhang et al performed a systematic review and meta analysis on safety and efficacy of intracavitary ECG for pediatric PICC insertion.
The manuscript is well written and the analysis well performed
English is correct, references are correct and figures and tables are relevant and well structured

Experimental design

The scope of the article is original and the sustematic review is well performed.
The standards for systematic reviews are met (GRADE) and the analysis was correctly registered (PROSPERO).
The methods are well described.

Validity of the findings

The impact of the paper is important, especially for neonatologist, that still do not use a lot this technique for the fear of technical failure.
In this regard I think it would be useful to separate the rates of safety and efficacy for the older children and for neonates, because as the authors correctly stated, it is sometimes difficult to have a correct P wave in neonatal PICCs due to the small caliber of the line.
Furthermore among the neonatal population it would be very usefull to have a subanalysis dividing the lines based on their caliber: 1 Fr vs 2 Fr catheters, to assess if it is feasible to perform intracavitary ekg in 1 Fr neonata lPICCs.
Conclusions are correct, but once agaim I would separate the rates of efficacy and safety for neonates and older children.

Additional comments

Regarding the statement about the fibrin sheath (line 245), I do not agree with the authors: it is the biofilm that forms inside the catheter and can be colonized over time. Fibrin sheath forms because the catheter is a foreign body, not because there is a damaged tissue.
It is difficult for me to understand why ekg decreases the number of first attempts: it should reduce the number of malpositions, thrombosis and phlebitis, but the insertion of the line is indipendent from the type of visualization of the tip... Do the authors have an explanation for this?

·

Basic reporting

- Language and Clarity: The manuscript generally uses clear and professional English throughout, making it accessible to a broad audience. However, some sections, particularly those discussing the distinction between neonates and the broader pediatric population, could benefit from more precise language to avoid any potential ambiguity.
- Literature References and Background: The manuscript is well-referenced, drawing on a substantial body of relevant literature. The background provides sufficient context for the study, effectively framing the importance of IC-ECG-guided PICC placement. However, it would be advantageous to further differentiate between the use of ECCs and PICCs in different pediatric age groups to strengthen the contextual grounding.
- Structure, Figures, and Tables: The article is professionally structured, adhering to the standard format expected for a systematic review and meta-analysis. Figures and tables are well-organized and effectively complement the text. The clarity in distinguishing between ECC and PICC in these visual elements could be improved to enhance the reader's understanding.
- Data Sharing: The authors have indicated that raw data or code from this study is not being submitted due to privacy and ethical restrictions. While this is understandable, it may limit the ability of other researchers to fully validate or replicate the findings. It is recommended that the authors provide a clear protocol for how interested researchers can request access to data for validation or collaboration purposes.
- Self-Contained and Relevant: The manuscript is self-contained, with all necessary information presented to support the hypotheses and conclusions drawn. The results are directly relevant to the research questions posed and are supported by the data provided.

Experimental design

- Relevance to Aims and Scope: The manuscript aligns well with the aims and scope of the journal, focusing on the efficacy and safety of IC-ECG-guided PICC placement in pediatric patients—a topic of significant clinical importance and relevance.
- Research Question: The research question is clearly defined, relevant, and meaningful. The study addresses a critical gap in the literature by systematically evaluating the use of IC-ECG-guided PICC placement in a pediatric population, which has not been comprehensively studied before. This focus on filling an identified knowledge gap is commendable.
- Rigor and Standards: The investigation appears to have been conducted to a high technical and ethical standard. The study design, including the inclusion of both randomized controlled trials and quasi-experimental studies, supports the rigor of the analysis. Ethical considerations, particularly in dealing with pediatric patients, seem to have been appropriately addressed.
- Methodological Detail and Reproducibility: The methods are described with sufficient detail, allowing for replication by other investigators. The systematic approach to literature search, study selection, and data extraction is well-documented. However, while the methodology is thorough, further clarification on certain aspects, such as the handling of heterogeneity in the meta-analysis, could enhance reproducibility. Providing more detailed information on how studies were assessed for quality and how potential biases were managed would strengthen the manuscript.
- Reproducibility: The methods should enable another researcher to reproduce the study, but there could be additional emphasis on describing the statistical methods used in more detail, particularly concerning the criteria for selecting fixed vs. random-effects models in the meta-analysis. This additional information would ensure that the study is fully reproducible by others.

Validity of the findings

- Impact and Novelty: The manuscript appropriately focuses on the efficacy and safety of IC-ECG-guided PICC placement in pediatric patients, a relevant and under-explored area. While the manuscript does not directly assess impact or novelty, it implicitly addresses these through its systematic review of existing studies. The encouragement of meaningful replication is well-founded, as the rationale and benefits to the literature are clearly articulated, particularly in highlighting the need for more standardized procedural techniques and further research into cost-effectiveness.
- Data Robustness and Statistical Soundness: The data provided in the manuscript appear robust, statistically sound, and well-controlled. The systematic review and meta-analysis methodology ensures that the results are based on a comprehensive and rigorous analysis of the existing literature. However, given that the authors have not provided raw data due to privacy concerns, the robustness of the conclusions relies heavily on the quality and reporting standards of the included studies.
- Conclusions and Link to Research Question: The conclusions are appropriately stated and closely linked to the original research question. The authors have limited their conclusions to those that are directly supported by the data, avoiding overreaching claims. However, it is important to note that while the manuscript presents correlations between IC-ECG-guided PICC placement and reduced complications, these findings should not be interpreted as establishing causative relationships. The conclusions should continue to emphasize that while IC-ECG is associated with improved outcomes, causality cannot be definitively established due to the observational nature of the included studies.
- Causality vs. Correlation: The manuscript correctly avoids making unwarranted causal claims, instead focusing on observed associations. The discussion of results should maintain this careful distinction, ensuring that readers understand the limitations of correlational findings, particularly in the context of the study designs included in the meta-analysis. The authors should be commended for adhering to this scientific rigor, but a reminder to explicitly state the limitations regarding causality would further strengthen the manuscript.

Additional comments

1. Distinction Between Pediatric Patients and Neonates:
- Comment: The manuscript frequently refers to "pediatric patients" as a broad category, which includes neonates. However, neonates have distinct physiological and clinical needs that differ significantly from children. I recommend that the authors explicitly differentiate between these groups where necessary to avoid potential misinterpretation of the findings.

2. Clarification on ECC and PICC Terminology:
- Comment: The manuscript seems to conflate the terms ECC (Epicutaneous Cava Catheter) and PICC (Peripherally Inserted Central Catheter). According to D'Andrea et al, a ECC is not an PICC, however the use of the term ECC is mainly found in Italian literature and specifically within the Journal of Vascular Access (JVA). The manuscript reflects some confusion in using these terms, which could lead to misunderstanding among readers: a ECC in neonatal research equals a PICC (what is in a name). I suggest the authors review their use of terminology, ensuring that they align with widely accepted definitions outside of the Italian context, to avoid any ambiguity.

3. Relevance of First Puncture Success Rate in Results:
- Comment: The manuscript discusses the "first puncture success rate" as a key result. However, the relevance of this metric in the context of the study's primary objectives is unclear, particularly in relation to PICC tip positioning using the IC-ECG method or compared to any other insertion method. The "first puncture success rate" typically relates to the initial venous access, but what does this have to do with the final PICC tip position, which is the critical outcome when using IC-ECG or other guided methods, such as expert ultrasound, that visualize and guide the tip during insertion? The authors should clarify whether they are referring to the successful insertion as correct PICC tip placement and explain the connection between the first puncture success rate and the accuracy of PICC tip placement using the IC-ECG method. This clarification would significantly improve the manuscript’s clarity and relevance.

---

## Round 0.2 · accepted · Accept

Dear Dr. Zhang,

Thank you for submitting the revised version of your manuscript. After a thorough review of the changes by the reviewers and myself, I am pleased to inform you that all the reviewers' comments have been adequately addressed. Therefore, your manuscript is ready for publication in PeerJ.

I thank all reviewers for their efforts in improving the manuscript and the authors' cooperation throughout the review process.

Sincerely yours,
Stefano Menini

·

Basic reporting

Well performed revision.

Experimental design

Revised accordingly

Validity of the findings

Revised accordingly

Additional comments

Revised accordingly